# Ubiquitin-Conjugating Enzymes Ubc1 and Ubc4 Mediate the Turnover of Hap4, a Master Regulator of Mitochondrial Biogenesis in *Saccharomyces cerevisiae*

**DOI:** 10.3390/microorganisms10122370

**Published:** 2022-11-30

**Authors:** Denise Capps, Arielle Hunter, Mengying Chiang, Tammy Pracheil, Zhengchang Liu

**Affiliations:** Department of Biological Sciences, University of New Orleans, New Orleans, LA 70148, USA

**Keywords:** *Saccharomyces cerevisiae*, Hap4, ubiquitin-conjugating enzyme (E2 enzyme), protein turnover, transcription factor, Ubc1, Ubc4, proteasome, mitochondrial dysfunction

## Abstract

Mitochondrial biogenesis is tightly regulated in response to extracellular and intracellular signals, thereby adapting yeast cells to changes in their environment. The Hap2/3/4/5 complex is a master transcriptional regulator of mitochondrial biogenesis in yeast. Hap4 is the regulatory subunit of the complex and exhibits increased expression when the Hap2/3/4/5 complex is activated. In cells grown under glucose derepression conditions, both the *HAP4* transcript level and Hap4 protein level are increased. As part of an inter-organellar signaling mechanism coordinating gene expression between the mitochondrial and nuclear genomes, the activity of the Hap2/3/4/5 complex is reduced in respiratory-deficient cells, such as *ρ*^0^ cells lacking mitochondrial DNA, as a result of reduced Hap4 protein levels. However, the underlying mechanism is unclear. Here, we show that reduced *HAP4* expression in *ρ*^0^ cells is mediated through both transcriptional and post-transcriptional mechanisms. We show that loss of mitochondrial DNA increases the turnover of Hap4, which requires the 26S proteasome and ubiquitin-conjugating enzymes Ubc1 and Ubc4. Stabilization of Hap4 in the *ubc1 ubc4* double mutant leads to increased expression of Hap2/3/4/5-target genes. Our results indicate that mitochondrial biogenesis in yeast is regulated by the functional state of mitochondria partly through ubiquitin/proteasome-dependent turnover of Hap4.

## 1. Introduction

Mitochondria are the power plants of eukaryotic cells and play important roles in intermediary metabolism. Except for a small number of respiratory chain components encoded in the mitochondrial genome, most mitochondrial proteins are encoded in the nuclear genome. Mitochondrial biogenesis requires coordinated expression of genes from both the nuclear and mitochondrial genomes [1,2]. Multiple signaling pathways between mitochondria and the nucleus exist to coordinate gene expression in the nucleus in response to changes in the functional state of mitochondria as well as changes in the cellular environment [3,4,5,6,7,8]. The prototypal pathway of this sort is the mitochondria-to-nucleus retrograde signaling pathway, also known as the RTG pathway. *Saccharomyces cerevisiae* maintains viability after losing mitochondrial DNA (mtDNA) and becomes a *ρ*^0^ “petite”. Mitochondrial dysfunction due to loss of mtDNA activates two basic helix-loop-helix leucine zipper transcription factors, Rtg1 and Rtg3, leading to changes in nuclear gene expression and a reconfiguration of metabolic pathways that adapt cells to mitochondrial defects [5]. Cells lacking their mitochondrial genome also induce the expression of *PDR5*, encoding an ATP-binding cassette transporter involved in multidrug resistance, through the activation of the zinc finger transcription factor Pdr3 [6,7].

The Hap2/3/4/5 transcription factor is a multi-subunit master regulator of mitochondrial biogenesis and an essential part of yeast’s metabolic remodeling when cells switch from glycolysis to respiratory metabolism for generating ATP [9,10,11,12,13]. The Hap2/3/5 trimer binds to CCAAT sequence elements in the promoters of target genes and requires Hap4 to provide transcriptional activation domain activity [14]. When yeast cells exhaust the supply of glucose, there is an increase in *HAP4* mRNA and Hap4 protein levels while the other members of the complex are constitutively expressed [12,14,15]. An elevation in Hap4 protein level results in the formation of the Hap2/3/4/5 complex to activate the expression of its target genes, including those encoding the enzymes of the tricarboxylic acid cycle and components of the mitochondrial respiratory complexes [13,16,17]. Apart from the Hap2/3/4/5 complex, a heme-activated protein, Hap1 also contributes to mitochondrial biogenesis by activating expression of genes under aerobic conditions, including those encoding components of Complexes III and IV of the electron transport chain [18]. 

Hap4’s importance in the regulation of the Hap2/3/4/5 complex and mitochondrial biogenesis has led to studies on the regulation of its expression. A *HAP4* promoter analysis reveals a 30-bp activating sequence at position −869 bp upstream of the ATG start codon that is important for its expression [19]. Cat8 is implicated in the regulation of *HAP4* expression by this activating sequence, but the effect has been proposed to be indirect. Zhang et al. reported that increased heme synthesis induces transcription of both *HAP4* and genes involved in respiratory metabolism [20]. They further showed that both Hap1 and Hap2 are positive regulators of *HAP4* expression in galactose-grown cells, suggesting that *HAP4* is subject to autoregulatory control. In glucose-grown cells, the role of Hap1 and Hap2 in *HAP4* expression is less clear: although *hap1* and *hap2* mutations increase *HAP4* expression in wild-type *HEM1* cells, they decrease *HAP4* expression in *hem1* mutant cells [20]. Rox1, a transcriptional repressor of hypoxic genes [21], negatively regulates *HAP4* expression in glucose-grown cells [20]. We recently reported that protein kinase A (PKA) is a potent inhibitor of *HAP4* expression. In mutants with reduced PKA activity such as *ira1*, *ira2*, *bcy1*, *gpb1 gpb2*, *pde1 pde2* mutants, *HAP4* expression is reduced. On the other hand, a complete lack of PKA activity due to a triple deletion mutation in the three genes encoding the catalytic subunits of PKA in a *yak1* mutant strain dramatically increases *HAP4* expression [22]. 

We have previously reported a transcriptional factor switch from the Rtg1/3 complex to the Hap2/3/4/5 complex for the expression of early tricarboxylic acid cycle genes, *CIT1*, *ACO1*, *IDH1*, and *IDH2*, in response to mitochondrial dysfunction [23]. In *ρ*^0^ cells, the expression of the rest of the tricarboxylic acid cycle is reduced, suggesting that the activity of the Hap2/3/4/5 complex is suppressed. Genome-wide transcriptional profiling reveals reduced expression of many Hap2/3/4/5-target genes as a result of reduced Hap4 protein level [24]. Other studies have shown that Hap4 is a highly unstable protein, whose turnover is affected by the cellular environment [25,26,27]. Elevated production of reactive oxygen species increases the turnover of Hap4 and consequently reduces mitochondrial biogenesis [25]. Conversely, an increase in the cellular heme level and glutathione redox state stabilizes Hap4, resulting in increased mitochondrial biogenesis [27]. Based on the findings that the *HAP4* transcript level does not change in *ρ*^0^ cells and that Hap4 overexpression leads to its localization in the vacuole, Bourges and colleagues have proposed that a vacuolar degradation mechanism might be behind reduced Hap4 protein levels in *ρ*^0^ cells [24]. It has also been suggested that reactive oxygen species may not be involved in the downregulation of Hap4 in response to mitochondrial dysfunction. 

In an aim to uncover the mechanism behind reduced Hap4 protein levels in *ρ*^0^ cells, we analyzed the expression of a *lacZ* reporter gene under the control of the *HAP4* promoter using β-galactosidase activity assays and Hap4 stability using a cycloheximide chase assay. We found that reduced Hap4 protein level in *ρ*^0^ cells is due to the combined effect of reduced promoter activity of the *HAP4* gene and increased turnover of Hap4 protein. We show that two ubiquitin-conjugating enzymes, Ubc1 and Ubc4, are required for Hap4 turnover. Our data suggest that ubiquitin-dependent turnover of Hap4 mediates mitochondria-to-nucleus signaling and plays a negative regulatory role in mitochondrial biogenesis.

## 2. Materials and Methods

### 2.1. Strains, Plasmids, and Growth Media and Growth Conditions

Yeast strains and plasmids used in this study are listed in Table 1 and Table 2, respectively. Yeast cells were grown at 30 °C in YNBcas5%D (0.67% yeast nitrogen base, 1% casamino acids, and 5% D-glucose), YNBcasR (0.67% yeast nitrogen base, 1% casamino acids, and 2% raffinose), YPGlycerol (1% bacto yeast extract, 2% peptone, and 3% glycerol [*v*/*v*]), and YPD (1% bacto yeast extract, 2% peptone, and 2% D-glucose). When required, uracil, adenine, and tryptophan were added to YNBcas5D and YNBcasR at standard concentrations to meet auxotrophic requirements [28]. The plate medium contains 2% agar. When indicated, 50 µM carbobenzoxyl-leucinyl-leucinyl-leucinal (MG132) was added to the growth media to inhibit proteasome activity. 

### 2.2. Creation of ρ^0^ Petites

To induce loss of mitochondrial DNA, *ρ*^+^ cells were grown to saturation in YPD liquid medium supplemented with 15 µg/mL ethidium bromide and then plated on YPD medium to obtain single colonies. Colonies were individually picked and grown overnight in YPD medium and cells were stained with DAPI as described [28]. *ρ*^0^ petites without mitochondrial DNA were confirmed by both fluorescence microscopy and inability to grow on YPGlycerol plate medium. 

### 2.3. Yeast Transformations and β-Galactosidase Activity Assays

Yeast strains were transformed with plasmids using the high-efficiency transformation method as described [28]. Cells were cultured in specified liquid medium overnight to reach OD_600_ 0.6~0.8 after undergoing a minimum of six cell divisions and collected for β-galactosidase assays using cellular extracts as described. For each plasmid-strain combination, β-galactosidase activities (in nanomoles of hydrolyzed o-nitrophenyl-β-D-galactopyranoside per milligram of protein per minute) were averages of duplicate assays of two to six independent cultures. Specific activity was calculated in relation to the total protein amount in the cellular extracts, which was determined using the Bradford assay with bovine serum albumin as the standard. When indicated, the means of β-galactosidase activities from two groups were compared by a *t*-test using Graphpad software (San Diego, CA, USA).

### 2.4. Cellular Extract Preparation and Immunoblotting

Total cellular proteins were prepared in extraction buffer (1.85 N NaOH-7.5% β-mercaptoethanol) followed by precipitation with trichloroacetic acid as described [33]. Trichloroacetic acid pellets were neutralized with 1M unbuffered Tris and resuspended in 1x SDS-PAGE loading buffer. Equivalent amounts of protein samples based on OD_600_ readings of the collected cell cultures were loaded into the lanes of the same gel. Immunoblotting of Hap4 with a C-terminal 3xHA epitope tag was carried out by incubating nitrocellulose membranes with rat monoclonal anti-HA antibody 3F10 (Roche Diagnostics GmbH, Mannheim, Germany), followed by goat anti-rat HRP-conjugated polyclonal secondary antibody (Jackson ImmunoResearch Laboratories, Inc., West Grove, PA, USA). For loading controls, blots were first deprobed in stripping buffer (2% sodium dodecyl sulfate, 100 mM β-mercaptoethanol, 62.5 mM Tris-HCl pH 6.7) for 45 min at 60 °C with agitation and then reprobed with anti-Ilv5 (acetohydroxyacid reductoisomerase) or anti-Pgk1 (3-phosphoglycerate kinase) rabbit polyclonal antibodies. Chemiluminescence images of Western blots were captured using the Bio-Rad Chemi-Doc photo documentation system (Bio-Rad Life Science, Hercules, CA, USA). A *t*-test was carried out to determine whether there was a significant difference in Hap4-HA protein levels from two groups of data. 

### 2.5. Cycloheximide Chase Assay

Cells expressing HA epitope-tagged Hap4 were grown in YNBcas5%D or YNBcasR medium to OD_600_ 0.6~0.8. Protein synthesis was then inhibited by the addition of 50 µg/mL cycloheximide to initiate chase. At 5 min intervals for a total of 15 min, 1 mL aliquots of cell cultures were withdrawn and immediately subject to cellular extract preparation. Immunoblotting of Hap4-HA was carried out as described above. For the MG132 treatment to inhibit the function of the 26S proteasomes, *erg6∆* mutants expressing *HAP4-HA* were grown in YNBcas5%D or YNBcasR medium overnight to reach OD_600_ ~0.4 and MG132 was added at a final concentration of 50 μM. Cycloheximide chase assays were then performed after cultures reached OD_600_ 0.6~0.8. For the determination of the half-life of Hap4-HA, band intensities on Western blots over the course of cycloheximide chase were quantified using the Bio-Rad QuantityOne software and used to fit an exponential curve, *y* = *Ae^−kt^*. The half-life was calculated from ln(2)/*k*. A *t*-test was carried out to determine whether there was a significant difference in the Hap4 half-lives from two groups of data.

## 3. Results

### 3.1. HAP4 Expression Is Affected by the Functional State of Mitochondria and the Quality of Carbon Sources

Yeast cells become *ρ*^0^ petites after losing mitochondrial DNA (mtDNA). It was reported that the Hap4 protein level is reduced by 2-fold in *ρ*^0^ cells grown in rich galactose media in comparison to that in *ρ*^+^ cells containing wild-type mtDNA [24]. Although it was suggested that transcriptional regulation was not involved, the underlying mechanism was unclear. To facilitate the analysis of transcriptional regulation of *HAP4*, we generated a *HAP4-lacZ* reporter gene by fusing a 1.8 kbp promoter sequence of *HAP4* to the *lacZ* coding sequence. The reporter gene was then introduced into otherwise wild-type *ρ*^+^ and *ρ*^0^ cells. Since Hap4 expression is also subject to transcriptional regulation in response to changes in carbon sources, transformants were grown in media with two different carbon sources, D-glucose (dextrose) and raffinose, and β-galactosidase assays on *HAP4-lacZ* expression were conducted. Raffinose is a fermentable trisaccharide composed of galactose, glucose, and fructose. When it is used as the sole carbon source in growth media, raffinose leads to glucose derepression and has been used extensively in the study of the mitochondria-to-nucleus retrograde signaling pathway [23,34,35]. Figure 1A shows that *HAP4-lacZ* expression is greatly increased in *ρ*^+^ cells grown in raffinose medium compared to dextrose, consistent with published results [12]. In *ρ*^0^ cells compared to *ρ*^+^ cells, the expression of the *HAP4-lacZ* reporter gene was reduced in both dextrose- and raffinose-grown cells, suggesting that *HAP4* transcription is reduced in response to mitochondrial DNA loss. 

We next examined Hap4 protein levels in *ρ*^+^ and *ρ*^0^ cells grown in dextrose and raffinose media. We generated a *HAP4-HA* fusion gene encoding a 3x human influenza hemagglutinin (HA) epitope tag at the C-terminus of Hap4. The fusion protein was determined to be functional by its ability to rescue the growth defect of a *hap4Δ* mutant in growth medium with nonfermentable carbon sources (Appendix A). When the protein level of Hap4-HA was determined by Western blotting, we found that Hap4 protein level is largely consistent with the promoter activity of *HAP4*: there is an increased Hap4 protein level in cells grown in raffinose medium compared to dextrose medium and a reduction in Hap4 protein level due to the loss of mtDNA (Figure 1B,C). However, the Hap4 protein level in *ρ*^0^ cells become largely independent of the carbon sources. The discrepancy between *HAP4-lacZ* reporter gene activities and Hap4-HA protein levels in *ρ*^0^ cells grown in dextrose versus raffinose medium suggest that a post-transcriptional mechanism(s) exists to modulate Hap4 protein levels.

### 3.2. Transcriptional Regulation of KGD1, a Target of the Hap2/3/4/5 Complex, Correlates with Protein Levels of Hap4

To help dissect the regulation of Hap4, we generated a *KGD1-lacZ* reporter gene as a readout of the activity of the Hap2/3/4/5 complex. *KGD1* encodes a subunit of the mitochondrial α-ketoglutarate dehydrogenase complex and is a target of the Hap2/3/4/5 complex [36]. Consistently, *KGD1-lacZ* has a low, basal expression in *ρ*^+^ cells grown in dextrose medium (glucose repression), and its expression is increased by 21-fold in *ρ*^+^ cells grown in raffinose medium (glucose derepression) (Figure 2A). *ρ*^0^ reduces *KGD1-lacZ* expression by 1.6- and 6.0-fold in dextrose- and raffinose-grown cells, respectively, consistent with reduced expression of Hap4 in *ρ*^0^ cells compared to *ρ*^+^ cells. In *ρ*^+^ cells grown in dextrose and raffinose media, *hap4Δ* reduces *KGD1*-lacZ expression by 2- and 6-fold, respectively. In contrast, *hap4Δ* has little to no effect on *KGD1-lacZ* expression in *ρ*^0^ cells grown in either dextrose or raffinose medium, suggesting that the Hap2/3/4/5 complex is largely inactive in *ρ*^0^ cells, likely due to a low expression level of Hap4. 

Overexpression of Hap4 leads to increased expression of Hap2/3/4/5-target genes [16,17,37]. Hap4 protein levels can be modulated by both the carbon sources and functional states of mitochondria. Mitochondria have many different functions. It is conceivable that mitochondrial biogenesis can be fine-tuned through different levels of Hap4. However, to our knowledge, this possibility has not been reported previously. We sought to examine *KGD1-lacZ* expression in cells expressing different levels of Hap4. Accordingly, we overexpressed Hap4 under the control of two strong heterologous promoters, *TEF2* and *GPD* (encoding translation elongation factor lα and glyceraldehyde-3-phosphate dehydrogenase, respectively) [38]. Plasmids encoding 3xHA epitope-tagged Hap4 under the control of the promoter of *TEF2*, *GPD*, or its own were introduced separately into a *hap4Δ* mutant strain carrying an integrated *KGD1-lacZ* reporter gene. Hap4-HA levels were detected by Western blotting. Figure 2B shows that in glucose-grown cells where *HAP4* is under the control of the heterologous promoter *TEF2* or *GPD* in comparison to its native promoter, Hap4 protein levels are much higher. Consistent with *GPD* being a stronger promoter than *TEF2* in dextrose-grown cells [38], the Hap4 protein level is also higher in cells expressing *GPDp-HAP4* than cells expressing *TEF2p-HAP4*. In raffinose-grown cells, *HAP4* expression under the control of either of the heterologous promoters is only marginally higher than its native promoter due to glucose derepression of *HAP4* expression.

We then determined *KGD1-lacZ* expression in cells expressing different levels of Hap4 via β-galactosidase activity assays. In dextrose-grown cells, overexpression of Hap4 under the control of the *GPD* or *TEF2* promoter increases *KGD1-lacZ* expression, and there is a positive correlation between Hap4 protein levels and *KGD1-lacZ* activity (Figure 2B,C). In raffinose-grown cells, the marginal overexpression of *HAP4* under the control of *GPD* and *TEF2* promoters slightly increases *KGD1-lacZ* expression. Significantly, in cells in which *HAP4* is under the control of the *GPD* or *TEF2* promoter, the expression of *KGD1* is significantly lower in cells grown in dextrose medium compared to raffinose medium even though Hap4-HA levels are comparable in those cells. This indicates that a regulatory mechanism other than the Hap4 protein level exists to achieve maximal induction of *KGD1-lacZ* expression under glucose derepression conditions. Together, these data support the notion that varying Hap4 protein levels lead to a graded transcriptional response of the target genes of the Hap2/3/4/5 complex, rather than an all-or-none effect. Since the Hap2/3/4/5 complex mediates the expression of many mitochondrial proteins [13,17,24], by modulating Hap4 protein levels, yeast cells are able to fine-tune mitochondrial biogenesis to meet changing energetic and metabolic requirements in cells.

### 3.3. Hap4 Has a Shorter Half-Life in ρ^0^ Cells than in ρ^+^ Cells

*HAP4* expression correlates with respiratory metabolism. Hap4 is needed when cells have functional mtDNA, which encodes several proteins essential for oxidative phosphorylation. When cells lose mitochondrial DNA and become unable to use nonfermentable carbon sources, it may be advantageous for cells to reduce the Hap4 protein level to down-regulate the expression of genes involved in respiratory metabolism. This is partly achieved by reducing the promoter activity of *HAP4* in *ρ*^0^ cells (Figure 1A). The discrepancy between the Hap4 protein level and the *HAP4-lacZ* reporter gene activity in *ρ*^0^ grown in raffinose medium in Figure 1 prompted us to conduct a comparative analysis on Hap4 stability in *ρ*^+^ and *ρ*^0^ cells. 

When we tried to transform *ρ*^0^ cells with a plasmid encoding *TEF2p-HAP4* or *GPDp-HAP4*, it was difficult to get transformants, suggesting that overexpression of *HAP4* in *ρ*^0^ cells is toxic. The fewer transformants we obtained may contain suppressor mutations. Nevertheless, we examined Hap4-HA expression in *ρ*^+^ and *ρ*^0^ cells expressing *GPDp-HAP4* by Western blotting. Figure 3A shows that the Hap4-HA protein level was significantly lower in *ρ*^0^ cells compared to *ρ*^+^ cells grown in both dextrose and raffinose medium. *GPD* encodes isozyme 3 of glyceraldehyde-3-phosphosphate dehydrogenase, a glycolytic enzyme. A genome-wide transcriptome analysis in response to mitochondrial dysfunctions by Epstein et al. shows that *ρ*^0^ cells slightly increase the expression of most genes encoding glycolytic enzymes, including *GPD* [39]. The metabolic reconfiguration in ATP production in *ρ*^0^ cells is expected since increased metabolic influx into glycolysis can compensate for the lack of ATP production from mitochondria. Together, these data suggest that loss of mitochondrial DNA may increase instability of Hap4-HA. 

We performed a cycloheximide chase assay to investigate whether Hap4 stability was different in *ρ*^+^ versus *ρ*^0^ cell. After the addition of cycloheximide to cell cultures to inhibit protein translation, aliquots of samples were collected every five minutes for a total of 15 min and Hap4 protein levels in cells during the time course were determined by Western blotting. We found that there is an increased turnover of Hap4-HA in *ρ*^0^ cells compared to *ρ*^+^ cell: In dextrose-grown cells, the half-life of Hap4-HA in *ρ*^+^ and *ρ*^0^ cells is 7.8 min and 4.5 min, respectively; in raffinose-grown cells, the half-life of Hap4-HA in *ρ*^+^ and *ρ*^0^ cells is 9.5 min and 3.4 min, respectively (Figure 3B–D). Faster turnover of Hap4 in *ρ*^0^ cells in comparison to *ρ*^+^ cells was also observed in cells expressing HA-tagged Hap4 from the *GPD* promoter (Appendix A). These data suggest that yeast cells can sense the functional state of mitochondria and regulate Hap4 stability accordingly. Together with data in Figure 1, our results suggest that a lower level of the Hap4-HA protein in *ρ*^0^ cells compared to *ρ*^+^ cells results from the combined effect of reduced transcription from the *HAP4* promoter and increased turnover of Hap4 protein.

### 3.4. Hap4 Turnover Requires the 26S Proteasome

Hap4 has been reported to be quickly turned over [25,26,27]. However, the mechanism behind Hap4 instability is not clear. A genome-wide analysis of ubiquitylated proteins uncovered Hap4 as a potential candidate [40]. To test whether Hap4-HA turnover requires the ubiquitin/proteasome system, we treated *HAP4-HA* expressing cells with the proteasome inhibitor MG132. An *erg6Δρ* mutant was used to facilitate the diffusion of MG132 into cells [41]. Figure 4A shows that MG132 treatment increases the stability of Hap4 in *ρ*^+^ and *ρ*^0^ cells grown in both dextrose and raffinose medium, suggesting that the proteasome/ubiquitin system is required for Hap4 turnover. We calculated the half-life of Hap4-HA in *erg6Δ* mutant cells without MG132 treatment and confirmed that there was also faster turnover of Hap4 in *ρ*^0^ cells in comparison to *ρ*^+^ cells grown in both dextrose medium and raffinose medium (Appendix A). The strains used to generate the data on Hap4-HA half-lives in Figure 3D and Appendix A are from different strain backgrounds, suggesting that increased Hap4-HA turnover in response to mitochondrial DNA loss is not strain specific.

In cells treated with MG132, we detected slower mobility forms of Hap4-HA on Western blots over a wide range, which are putative ubiquitinated forms of Hap4-HA based on the migration pattern (Figure 4B). Faster turnover of Hap4-HA in *ρ*^0^ cells compared to *ρ*^+^ cells suggest that there may be increased ubiquitination of Hap4 in response to loss of mitochondrial DNA. To test this possibility, we compared Hap4-HA on Western blot from protein samples generated from *ρ*^+^ and *ρ*^0^
*erg6Δ* mutant cells grown in raffinose medium and treated without or with MG132 for 4 hours. We loaded 2.3 times as much protein extracts from *ρ*^0^ cells as from *ρ*^+^ cells onto the gel to have comparable levels of non-ubiquitinated Hap4. Figure 4B shows that there is increased ubiquitination of Hap4-HA in *ρ*^0^ cells compared to *ρ*^+^ cells. Together, our data suggest that faster turnover of Hap4 in *ρ*^0^ cells results from its increased ubiquitination. 

The function of the 26S proteasome is essential for cell viability [42]. *erg6Δ* mutant cells treated with 50 μM MG132 are slow-growing but still viable due to the residual activity of the proteasome (20–30% full activity) [41]. Therefore, it is not surprising that MG132 treatment does not completely abolish the turnover of Hap4-HA, especially in *ρ*^0^ cells (Figure 4A). Although the effect of MG132 on Hap4-HA stability was variable, we observed that MG132 was least effective in stabilizing Hap4-HA in *ρ*^0^ cells grown in dextrose medium (Figure 4A). This may be explained by increased expression of *PDR5* in *ρ*^0^ cells grown in dextrose medium [43]. Pdr5 is a ATP-binding cassette drug efflux pump that reduces the efficacy of proteasome inhibitors [44]. It is also likely that Hap4 may be subjected to proteosome-independent turnover in *ρ*^0^ cells. To differentiate these two possibilities, we examined Hap4 stability in a *pre1 pre4* double mutant, which is defective in the proteasomal function [29]. Figure 4C shows that the *pre1 pre4* double mutation increases the stability of Hap4-HA to a similar extent in *ρ*^+^ and *ρ*^0^ cells grown in dextrose medium and in *ρ*^+^ cells grown in raffinose medium. *ρ*^0^
*ubc1 ubc4* mutant cells did not grow in raffinose medium and thus were not included in the analysis in Figure 4C. Altogether, our data suggest that Hap4 turnover requires the 26S proteasome and loss of mitochondrial DNA increases Hap4 ubiquitination and turnover. 

### 3.5. Hap4-HA Is Stabilized in a ubc1Δ ubc4Δ Double Mutant

Protein ubiquitination requires the activity of a cascade of three enzymes: an E1 ubiquitin-activating enzyme, an E2 ubiquitin-conjugating enzyme (Ubc), and an E3 ubiquitin ligase [45,46]. The yeast genome encodes a family of 13 ubiquitin-conjugating enzymes, two of which, Ubc3/Cdc34 and Ubc9, are essential [42]. Despite its sequence similarity to other ubiquitin-conjugating enzymes, Ubc9 is a SUMO-conjugating enzyme [47]. To identify the E2 enzyme(s) responsible for Hap4 turnover, Hap4-HA stability was examined using cycloheximide chase in 11 mutants each carrying a deletion mutation of a non-essential *UBC* gene as well as two mutants carrying temperature-sensitive alleles of *CDC34* or *UBC9*. Since Hap4-HA is least stable in *ρ*^0^ cells grown in raffinose medium (Figure 3D), we generated *ρ*^0^ derivatives of the 13 *ubc* single mutants and grew them in raffinose medium to test Hap4-HA stability to maximize the chance of identifying the E2 enzyme responsible for Hap4 turnover. Additionally, *cdc34-2* and *ubc9-1* mutant cells were switched to 37 °C, a non-permissive temperature for both mutants, before cycloheximide chase was initiated. Figure 5A,B show that Hap4-HA is somewhat stabilized in *ubc1* and *ubc5* single deletion mutants. Ubc1, 4 and 5 are known to have partial overlapping functions and constitute an enzyme sub-family that is essential for cell growth and viability [48,49]. To test whether these three enzymes have a redundant function in mediating Hap4 turnover, we determined Hap4-HA stability using cycloheximide chase assay in *ρ*^0^ derivatives of *ubc1/4Δ*, *ubc1/5Δ*, and *ubc4/5Δ* double mutants grown in raffinose medium. Figure 5B shows that *ubc1/4Δ* significantly increases Hap4-HA stability. *ubc4/5Δ* seems to slightly increase Hap4-HA stability. Hap4-HA stability in *ubc1/5Δ* mutant cells is similar to what is observed in the *ubc1Δ* single mutant. We then determined Hap4-HA stability in *ubc1/4Δ ρ*^+^ cells grown in raffinose medium and *ubc1/4Δ ρ*^+^ and *ρ*^0^ cells grown in dextrose medium. We found that Hap4-HA also shows increased stability under these conditions (Figure 5C,D). Together, our data indicate that Ubc1 and Ubc4 are the primary ubiquitin-conjugating enzymes that mediate Hap4 turnover. Surprisingly, quantitative analysis of Hap4-HA stability shows that there is no significant difference in its half-lives between *ρ*^+^ and *ρ*^0^ cells of the wild type Y0002 strain grown in dextrose medium (Figure 5D, left panel and Appendix A). Nevertheless, Hap4-HA is still less stable in *ρ*^0^ cells than *ρ*^+^ cells of the Y0002 strain when they are grown in raffinose medium (Figure 5D, right panel and Appendix A). The strains used to generate the data in Figure 5C,D are from Y0002 background, which is different from PSY142 background for generating the data in Figure 3D and BY4741 background in Appendix A. A comparison of Hap4-HA stability in these three strains shows that *ρ*^0^ cells have similar Hap4 half-lives in dextrose- and raffinose-grown cells, respectively (Figure 3D, Appendix A). The lack of a difference in Hap4-HA half-lives between Y0002 *ρ*^+^ and *ρ*^0^ cells grown in dextrose medium appears to be due to its increased turnover in *ρ*^+^ cells. *ρ*^+^ Y0002 strain grown in raffinose medium also shows increased turnover of Hap4-HA compared to *ρ*^+^ cells of PSY142 and BY4741 background strains grown in the same medium. We hypothesize that an unknown mitochondrial defect in Y0002 cells might account for increased Hap4 instability in *ρ*^+^ cells, especially in cells grown in dextrose medium. It is also possible that other strain background differences might be responsible.

We observed slower mobility forms of Hap4-HA on Western blots from protein samples of *erg6* mutant cells treated with MG132 to inhibit the proteasome function (Figure 4B and Appendix A). They are putative ubiquitinated forms of Hap4-HA. Consistently, these slower mobility forms are also observed in *pre1 pre4* mutant cells with reduced proteasomal function, without MG132 treatment (Appendix A). Our data in Figure 5 suggest that Ubc1 and Ubc4 are primary ubiquitin-conjugating enzymes responsible Hap4-HA turnover. A double mutation in *UBC1* and *UBC4* is expected to reduce ubiquitination of Hap4-HA. To test this possibility, we generated an *erg6 ubc1 ubc4* triple mutant, grew cells in the presence of MG132, and examined Hap4-HA by Western blotting. Appendix A shows that slower mobility forms of putative ubiquitinated Hap4-HA are barely detectable. Our data suggest that stability of Hap4-HA can be increased by reducing either the cellular proteasomal function or Hap4 ubiquitination. 

### 3.6. Hap4 Stabilization Due to ubc1Δ ubc4Δ Increases Expression of Hap2/3/4/5-Target Genes

We next asked whether Hap4 turnover via the ubiquitin/proteasome system impacts the activity of the Hap2/3/4/5 complex. To this end, the expression of the *lacZ* reporter gene under the control of the promoter of *KGD1* or *SDH1* was analyzed in wild type, *hap4Δ* single mutant, *ubc1/4Δ* double mutant, and *ubc1/4Δ hap4Δ* triple mutant cells (Figure 6). *SDH1* encodes the A subunit of the succinate dehydrogenase complex and is another target gene of the Hap2/3/4/5 complex [13,17,24]. In *ρ*^+^ cells grown in dextrose medium, *ubc1/4Δ* significantly increases the expression of these two reporter genes, and these increases are largely reversed in the *ubc1/4Δ hap4Δ* triple mutant (Figure 6A). Similar results were obtained in *ρ*^+^ cells grown in raffinose medium (Figure 6B). Importantly, *hap4Δ* completely abolishes increased expression of *KGD1-lacZ* and *SDH1-lacZ* reporter genes in *ubc1/4Δ* mutant cells grown in raffinose medium. These data indicate that Hap4 turnover mediated by Ubc1 and Ubc4 reduces the activity of the Hap2/3/4/5 complex. 

## 4. Discussion

Loss of mitochondrial DNA in yeast cells leads to reduced mitochondrial biogenesis by downregulating the protein level of Hap4. Here, we show that this downregulation is due to the combined effect of reduced transcription at the *HAP4* promoter and increased turnover of the Hap4 protein. The mitochondrial genome encodes several key components of mitochondrial respiratory complexes [50]. Without mitochondrial DNA, cells cannot undergo oxidative phosphorylation. Many mitochondrial proteins involved in the tricarboxylic acid cycle and the electron transport chain are encoded in the nuclear genome and are under the control of the Hap2/3/4/5 complex. Regulation of the activity of Hap2/3/4/5 complex via modulating Hap4 expression helps to achieve mitochondrial homeostasis by coordinating gene expression from the nuclear and mitochondrial genomes. Our study reveals a negative regulatory mechanism of Hap4 through its degradation via the ubiquitin/proteasome system and provides important insights into the regulation of this important inter-organellar signaling process.

Our data is the first to describe ubiquitin/proteasome-dependent turnover of Hap4. We identified two ubiquitin-conjugating enzymes, Ubc1 and Ubc4, that are responsible for fast Hap4 turnover. Our findings echo the role of these two enzymes in the ubiquitination of fructose-1,6-bisphosphatase as well as substrates targeted by anaphase promoting complex (APC), a multi-subunit E3 ubiquitin ligase [51,52]. Ubc1, Ubc4, and Ubc5 have a redundant essential function [48,53]. All three are required for selective protein degradation, while Ubc1 seems to play a special role in the early stages of growth upon spore germination. Ubc4 and Ubc5 are closely related, with 94% sequence identity. These two enzymes and their orthologs from other species comprise the largest subfamily of E2 enzymes [54]. Therefore, it was surprising not to observe significant stabilization of Hap4 in the *ubc1 ubc5* double mutant initially (Figure 5B). This is likely to be due to a higher expression level of Ubc4 compared to Ubc5. In a global analysis of protein expression in yeast, Ubc4 was found to be much more abundant than Ubc5 [55]. Consistently, a *ubc1 ubc4* double mutant grows much slower than a *ubc1 ubc5* double mutant [32]. Thus, in the *ubc1 ubc4* double mutant, the lower level of Ubc5 may not be sufficient to carry out Hap4 ubiquitination and degradation. Nevertheless, it’s clear that Ubc1 and Ubc4 play a redundant role in Hap4 turnover. Ubc1, Ubc4, and Ubc5 are known to work with two E3 ubiquitin ligases, Rsp5 and APC [49,51]. However, we failed to detect Hap4 stabilization in an *rsp5* mutant as well as in a mutant with a deletion mutation of *DOC1*, encoding a component of APC (our unpublished result).

Bourges et al. was unable to detect the signal of GFP-tagged Hap4 in cells with a single genomic copy using fluorescence microscopy [24]. When Hap4 was overexpressed, they found that Hap4-GFP was localized in the vacuole. Most of the data presented herein regarding the expression level and stability of Hap4 involved the use of HA-tagged Hap4 under the control of its own promoter. We did not detect vacuolar localization of GFP-tagged Hap4 under the control of its own promoter (our unpublished result). Whether vacuolar localization of Hap4 is an artifact of Hap4 overexpression is not a focus of this study. Future studies will be needed to determine potential contribution of the vacuolar degradation pathway to the regulation of Hap4.

Hap4 is stabilized in both *ρ*^+^ and *ρ*^0^
*ubc1 ubc4* double mutants grown in either dextrose or raffinose medium (Figure 5). Therefore, proteasome-mediated Hap4 turnover seems to also play a housekeeping role by suppressing the activity of the Hap2/3/4/5 complex in *ρ*^+^ cells. Respiratory metabolism generates reactive oxygen species and oxidative stress [56,57]. Ubiquitin in yeast is encoded by four different genes, three of which form fusion genes with those encoding ribosomal proteins, and the fourth is *UBI4*, encoding a polyubiquitin chain comprising five head-to-tail repeats [58]. *UBI4* expression is subject to glucose derepression mainly through the activation of the Hap2/3/4/5 complex and contributes to oxidative stress resistance in respiratory yeast cells [59,60]. Hap2/3/4/5-dependent expression of Ubi4 and ubiquitin-mediated Hap4 degradation may constitute a feedback control circuit to put a brake on respiratory metabolism. Many transcription factors are short-lived, which allows cells to respond quickly to changes in their environment. Ubiquitin-dependent degradation of transcription factors generally serves as a mechanism in down-regulating their activities. However, in some cases, ubiquitin-mediated turnover of transcription factors has been reported to play a stimulatory role in target gene expression [31]. This is unlikely to be the case for Hap4 since stabilization of Hap4 due to a *ubc1 ubc4* double mutation increases the expression of Hap2/3/4/5-target genes (Figure 6).

What is the signal in *ρ*^0^ cells that leads to reduced Hap4 expression? This is unlikely to be the loss of mitochondrial DNA, per se. In a genome-wide gene expression analysis using various respiratory-deficient mutants as well as *ρ*^0^ cells, the expression of Hap2/3/4/5-target genes is reduced in all of these mutants [24]. In *ρ*^0^ cells and in an *oxa1Δ* mutant, which have defects in the assembly and function of three respiratory complexes, reduced expression of Hap2/3/4/5-target genes is greater compared to other respiratory-deficient mutants in which only one respiratory complex is affected. Therefore, it is likely that more than one signal in *ρ*^0^ cells may regulate Hap4 expression. Hap4 turnover has been reported to be under the control of the redox environment in cells. For example, increased production of reactive oxygen species in a *tpk3Δ* mutant grown in minimal lactate medium reduces the protein level of Hap4; conversely, an increased glutathione redox state due to cAMP treatment in a *pde2Δ* mutant grown in minimal lactate medium stabilizes Hap4 [25,27]. Since *ρ*^0^ cells have been reported to have reduced levels of reactive oxygen species and exhibit normal expression of oxidative stress responsive genes [24,61], it remains to be determined if reactive oxygen species play a role in reducing *HAP4* expression in response to mitochondrial dysfunction.

The mechanism behind reduced Hap4 expression in *ρ*^0^ cells is expected to be different from that underlying intergenomic signaling, in which a set of nuclear genes are down-regulated in *ρ*^0^ cells but not in nuclear *pet* mutant *ρ*^+^ cells that possess mtDNA but lack respiration [3]. In the mitochondria-to-nucleus retrograde signaling pathway, we proposed that ATP might be a signaling molecule by mediating the interaction between Rtg2 and Mks1, which is a positive and negative regulator of this pathway, respectively [5,62]. Rtg2 binding to Mks1 leads to activation of retrograde signaling. Conversely, Mks1 dissociation from Rtg2 inhibits the two transcriptional activators of the pathway, Rtg1 and Rtg3. ATP, used at physiological concentrations, can dissociate Mks1 from Rtg2 from three different fungal species in an in vitro assay. Unlike vacuolar proteases, the ubiquitin/proteasome-dependent degradation of proteins is ATP-dependent [63]. *ρ*^0^ cells have similar levels of ATP to that of *ρ*^+^ cells when grown in dextrose medium and much lower ATP levels than *ρ*^+^ cells when glucose is exhausted [64,65]. Increased turnover of Hap4 in *ρ*^0^ cells is thus unlikely to be the result of direct ATP sensing via the ubiquitin/proteasome system. Future work will be directed toward the elucidation of the mechanisms underlying Hap4 turnover and transcriptional regulation of *HAP4*. Identification and characterization of the involved protein factors will help to understand how mitochondrial homeostasis is achieved through coordinated expression of genes encoding mitochondrial proteins from the mitochondrial and nuclear genomes, a process central to the growth and metabolism in eukaryotic cells. 

## Figures and Tables

**Figure 1 microorganisms-10-02370-f001:**
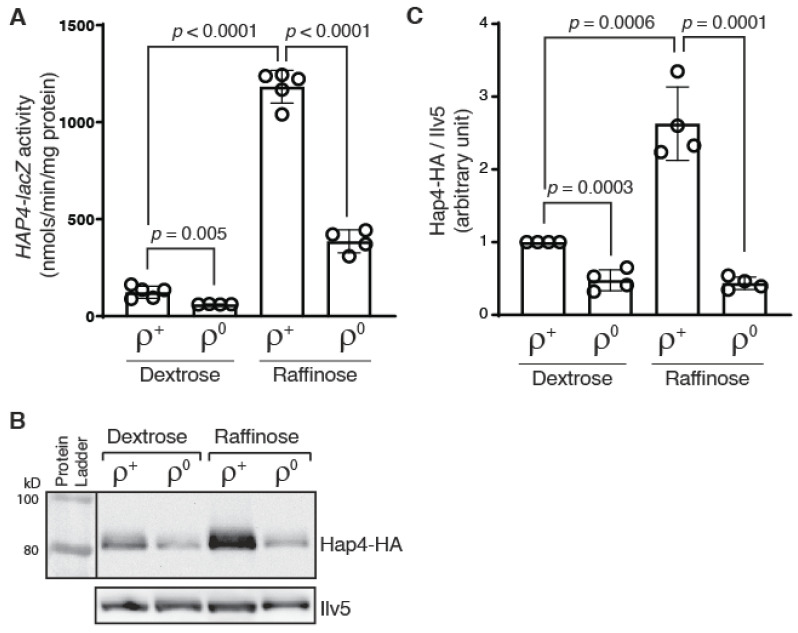
*HAP4* expression is reduced in *ρ*^0^ cells. (**A**) A β-galactosidase activity assay of the expression of a *HAP4-lacZ* reporter gene in wild-type *ρ*^+^ strain (BY4741) and its *ρ*^0^ derivative grown in dextrose and raffinose media. β-galactosidase activities were determined as described in Materials and Methods. The data is presented as the mean ± standard deviation. A *t*-test was carried out and the *p* values indicate a significant difference between two groups of data. (**B**) Immunoblotting of Hap4-HA in a *ρ*^+^
*hap4Δ* mutant (BY4741 *hap4*) and its *ρ*^0^ derivative grown in dextrose and raffinose media. Equivalent amounts of protein samples based on OD_600_ readings of the collected cell cultures were loaded into the lanes of the same gel. Ilv5 was included as a loading control. The result was representative of four independent sets of results. (**C**) Quantification of Hap4-HA/Ilv5 from Western blotting data.

**Figure 2 microorganisms-10-02370-f002:**
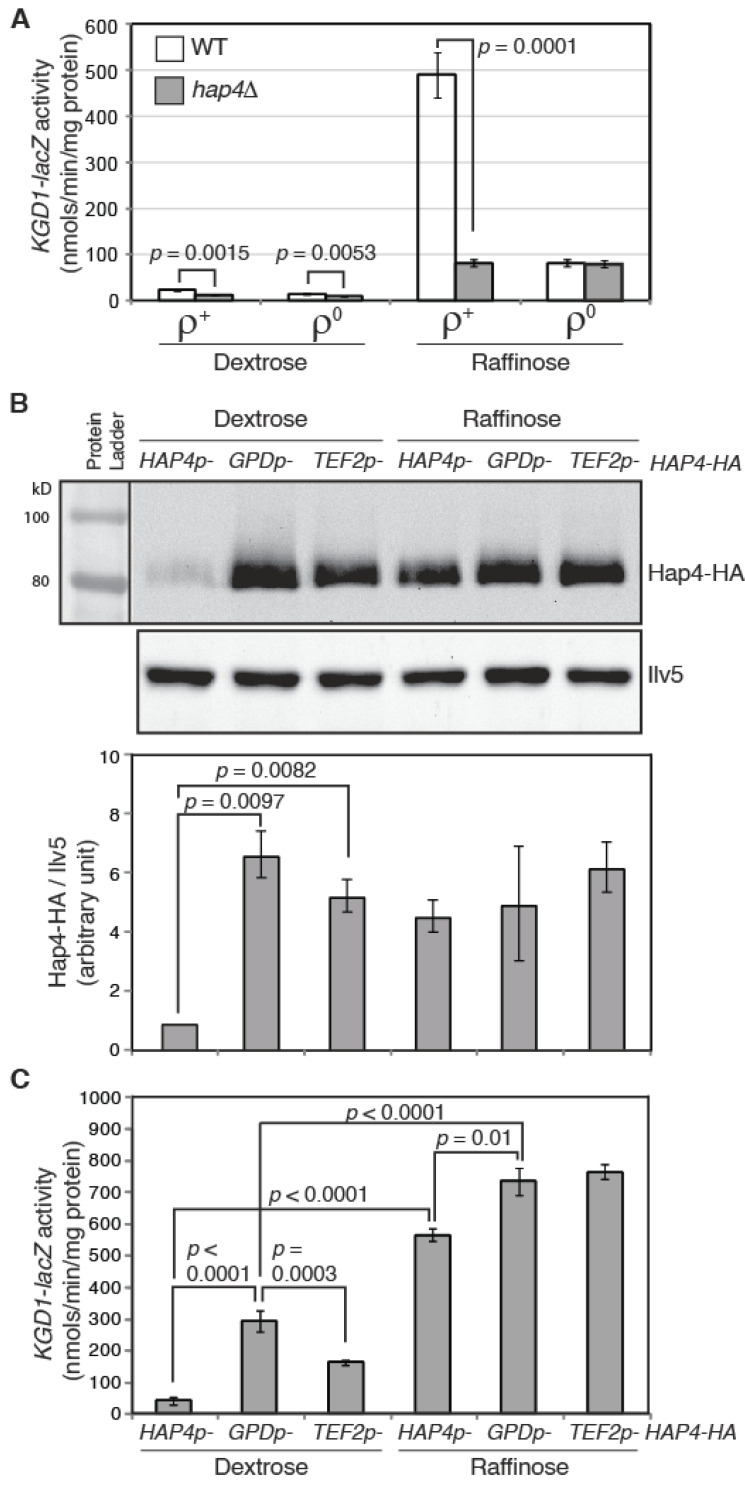
Expression of *KGD1-lacZ* reporter gene correlates with Hap4-HA protein levels. (**A**) A β-galactosidase activity assay of *KGD1-lacZ* expression in *ρ*^+^ and *ρ*^0^ cells of wild-type strain (ZLY3440) and its isogenic *hap4Δ* mutant (DCY247). (**B**) (upper panel) Immunoblotting of Hap4-HA in *hap4Δ* cells (DCY247) carrying centromeric plasmids encoding *HAP4-HA* under the control of its endogenous promoter *HAP4p*, a heterologous promoter *GPDp or TEF2p*. Equivalent amounts of protein samples based on OD_600_ readings of the collected cell cultures were loaded into the lanes of the same gel. Ilv5 was included as a loading control. (lower panel) Quantification of Hap4-HA/Ilv5 from two sets of Western blotting results. (**C**) A β-galactosidase activity assay of *KGD1-lacZ* expression in *hap4Δ* mutant cells carrying plasmids as described for panel (**B**).

**Figure 3 microorganisms-10-02370-f003:**
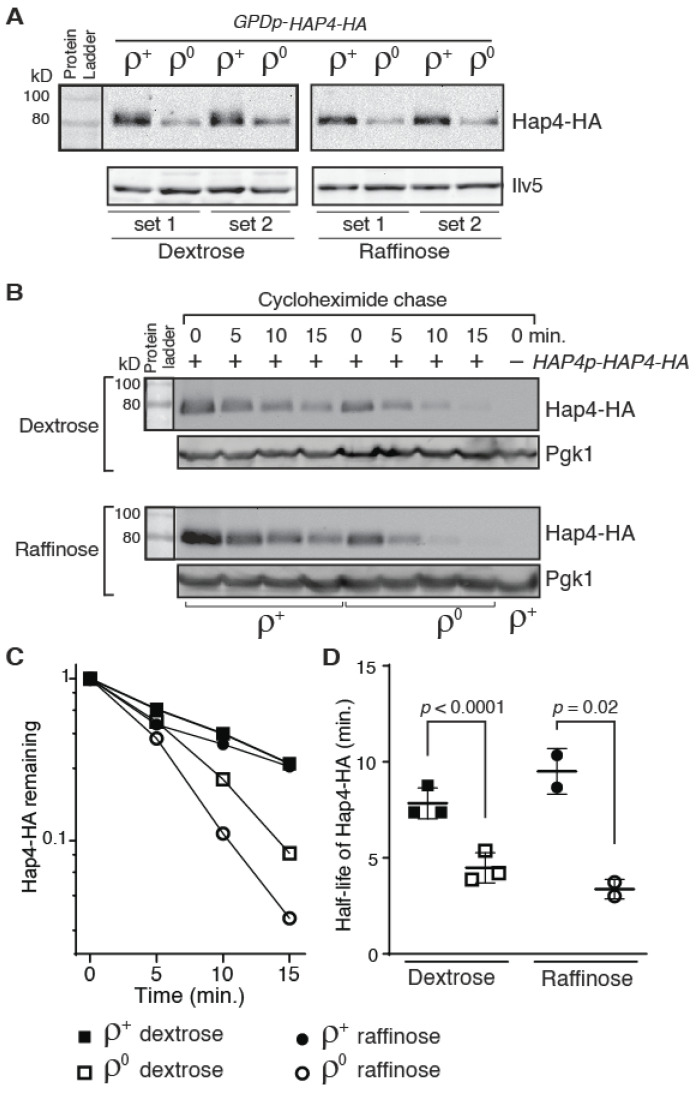
Hap4-HA is less stable in *ρ*^0^ cells than in *ρ*^+^ cells. (**A**) Western blot analysis of Hap4-HA expression in *hap4Δ* mutant cells (DCY247) carrying a centromeric plasmid encoding *GPDp-HAP4-HA* (pDC216). Equivalent amounts of protein samples based on OD_600_ readings of the collected cell cultures were loaded into the lanes of the same gel. (**B**) A cycloheximide chase assay of Hap4-HA stability in *ρ*^+^ and *ρ*^0^ cells of strain DCY247 grown in dextrose and raffinose media. Expression of *HAP4-HA* was under the control of its own promoter and the cycloheximide chase assay was conducted as described in Materials and Methods. Pgk1 was included as a loading control. (**C**) Quantification of Hap4-HA levels in panel (**B**). (**D**) The half-lives of Hap4-HA were determined from Western blots of Hap4-HA from *ρ*^+^ and *ρ*^0^ cells as described for panel (**B**) and plotted in the graph.

**Figure 4 microorganisms-10-02370-f004:**
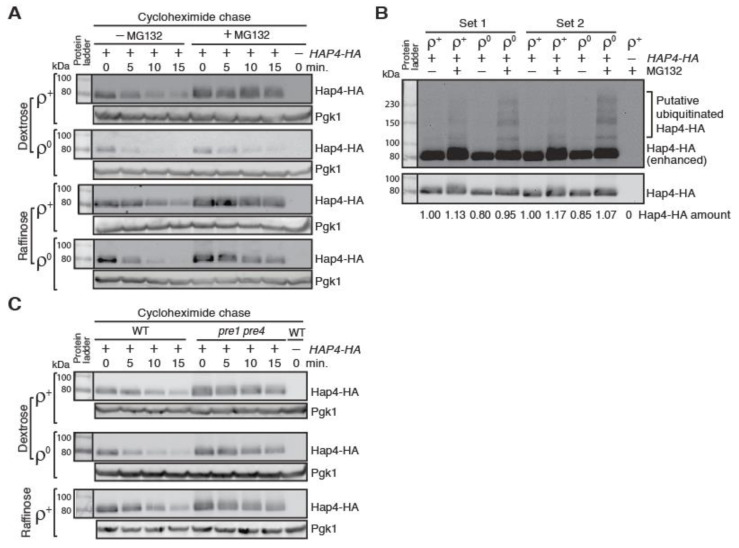
Inhibition of the activity of the 26S proteasome leads to increased stability of Hap4-HA. (**A**) A cycloheximide chase assay on Hap4-HA in *ρ*^+^ and *ρ*^0^ cells of an *erg6Δ* mutant strain (ZLY4531) with or without the treatment of the proteasome inhibitor MG132. Hap4-HA was detected using immunoblotting. Pgk1 was a loading control. (**B**) Increased formation of slower mobility forms of Hap4-HA in *ρ*^0^ cells compared to *ρ*^+^ cells when the proteasomal function is inhibited. *ρ*^+^ and *ρ*^0^ cells of an *erg6Δ* mutant strain expressing *HAP4-HA* from its own promoter were grown in raffinose medium with or without MG132 treatment. Protein samples were separated by SDS-PAGE and Hap4-HA was detected by Western blotting. The gel picture in the upper panel was enhanced to visualize the putative ubiquitinated forms of Hap4-HA. (**C**) Hap4-HA is stabilized in a *pre1 pre4* mutant, which is defective in proteasome protease function. *ρ*^+^ and *ρ*^0^ cells of wild-type strain (15Daub) and its isogenic *pre1 pre4* double mutant (PY555) were grown in dextrose or raffinose medium as indicated. Hap4-HA was detected by immunoblotting. Pgk1 was a loading control.

**Figure 5 microorganisms-10-02370-f005:**
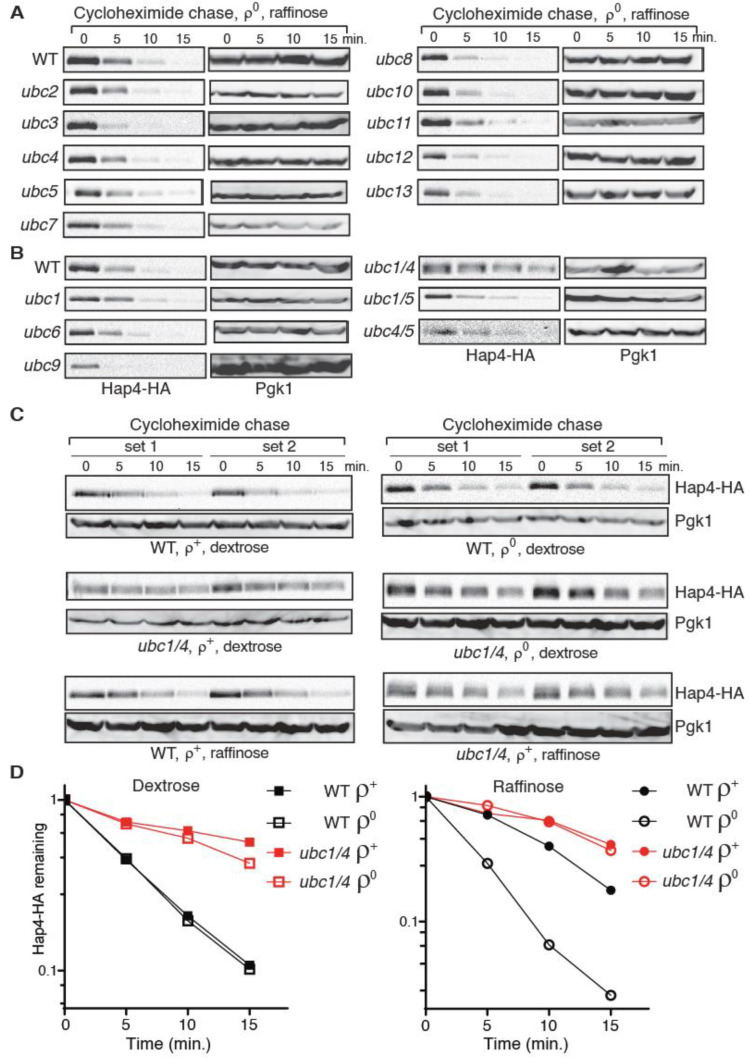
A *ubc1Δ ubc4Δ* double mutation stabilizes Hap4-HA. (**A**,**B**) A cycloheximide chase assay on Hap4-HA stability in *ρ*^0^ cells of wild type (BY4741 in panel A; Y0002 in panel B) and respective isogenic *ubc* mutant strains as indicated grown in raffinose medium. Hap4-HA was detected using Western blotting. (**C**) Hap4-HA is stabilized in *ρ*^+^ and *ρ*^0^ cells of a *ubc1/4Δ* mutant (ZLY3359) grown in dextrose or raffinose medium as indicated. (**D**) Quantification of Hap4-HA levels from *ρ*^+^ and *ρ*^0^ cells of wild type (WT, Y0002) and isogenic *ubc1/4Δ* mutant (ZLY3359) grown in dextrose and raffinose medium over the cycloheximide chase period. The means of remaining Hap4-HA levels during the chase period from two independent sets of experiments were plotted in the graphs.

**Figure 6 microorganisms-10-02370-f006:**
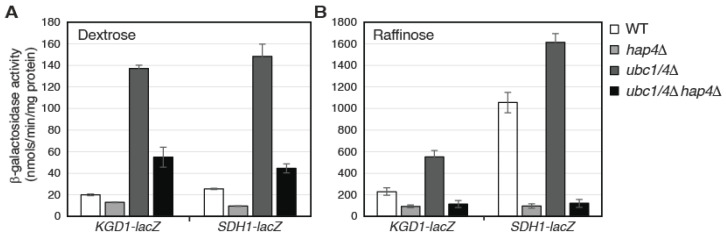
Hap4 stabilization in *ubc1Δ ubc4Δ* double mutant cells results in increased expression of Hap2/3/4/5-target genes. β-galactosidase assays on the expression of *KGD1-lacZ* and *SDH1-lacZ* reporter genes in wild-type (WT, Y0002) and isogenic mutant strains (*hap4Δ*, AHY145; *ubc1/4Δ*, ZLY3359; *ubc1/4Δ hap4Δ*, ZLY3839) grown in dextrose (**A**) and raffinose media (**B**).

**Table 1 microorganisms-10-02370-t001:** *S. cerevisiae* strains used in this study.

Strain	Genotype	Source	Application
BY4741	*MATa ura3 leu2 his3 met15*	ATCC	Figures 1A, 5A and S1
BY4741 *hap4*	BY4741 *hap4::kanMX4*	Yeast genome deletion project (YGDP)	Figures 1B and S1
PSY142	*MATα ura3 leu2 lys2*	Liu Lab stock	
DCY210	PSY142 *KGD1-lacZ::URA3*	This study	
ZLY3440	PSY142 *KGD1-lacZ::ura3::kanMX4*	This study	Figure 2
DCY247	PSY142 *KGD1-lacZ::ura3 hap4::LYS2*	This study	Figures 2, 3 and S2
ZLY4531	BY4741 *erg6::kanMX4*	This study	Figures 4A, 4B, S3 and S5A
15Daub	*MATa ura3Dns ade1 his2 leu2-3,112 trp1-1 bar1D*	[29,30]	Figures 4C and S5B
PY555	*MATa ura3Dns ade1 his2 leu2-3,112 trp1-1 pre1-1 pre4-1*	[29]	Figures 4C and S5B
BY4741 ubc2	BY4741 *ubc2::kanMX4*	YGDP	Figure 5A
BY4741 ubc4	BY4741 *ubc4::kanMX4*	YGDP	Figure 5A
BY4741 ubc5	BY4741 *ubc5::kanMX4*	YGDP	Figure 5A
BY4741 ubc7	BY4741 *ubc7::kanMX4*	YGDP	Figure 5A
BY4741 ubc8	BY4741 *ubc8::kanMX4*	YGDP	Figure 5A
BY4741 ubc10	BY4741 *ubc10::kanMX4*	YGDP	Figure 5A
BY4741 ubc11	BY4741 *ubc11::kanMX4*	YGDP	Figure 5A
BY4741 ubc12	BY4741 *ubc12::kanMX4*	YGDP	Figure 5A
BY4741 ubc13	BY4741 *ubc13::kanMX4*	YGDP	Figure 5A
RJD2141 (cdc34-2)	*MATa ura3 leu2 his3 GCN4 (Myc9)::HIS3 cdc34-2*	[31]	Figure 5A
Y0002	*MATα his3-Δ200 leu2-3,2-112 lys2-801 trp1-1(am) ura3-52*	[32]	Figures 5B, 6 and S4
Y0151 (ubc1)	Y0002 *ubc1::HIS3*	[32]	Figure 5B
Y0026 (ubc6)	Y0002 *ubc6::HIS3*	[32]	Figure 5B
Y0233 (ubc9-1)	Y0002 *ubc9Δ::TRP1 ubc9-1::LEU2 bar1::HIS3*	[32]	Figure 5B
Y0501 (ubc1/5)	Y0002 *ubc1::URA3 ubc5::LEU2*	[32]	
Y0096 (ubc4/5)	Y0002 *ubc4::HIS3 ubc5::LEU2*	[32]	Figure 5B
Y0108 (ubc1/4)	Y0002 *ubc1::URA3 ubc4::HIS3*	[32]	
ZLY3356 (ubc1/5)	Y0002 *ubc1::ura3::kanMX4 ubc5::LEU2*	This study	Figure 5B
ZLY3359 (ubc1/4)	Y0002 *ubc1::ura3::kanMX4 ubc4::HIS3*	This study	Figures 5B,C and 6
ZLY4636 (erg6 ubc1/4)	Y0002 *ubc1::ura3::kanMX4 ubc4::HIS3 erg6::kanMX4*	This study	Figure S5C
AHY145	Y0002 *hap4::LYS2*	This study	Figure 6
Y0003	*MATa his3-Δ200 leu2-3,2-112 lys2-801 trp1-1(am) ura3-52*	[32]	
TPY1277	Y0003 *hap4::kanMX4* (for making ZLY3839 via tetrad analysis)	This study	
ZLY3839	Y0002 *ubc1::ura3::kanMX4 ubc4::HIS3 hap4::KanMX4*	This study	Figure 6

**Table 2 microorganisms-10-02370-t002:** Plasmids used in this study.

Plasmid	Description	Source	Application
pDC124	pRS416-HAP4-lacZ, expressing *lacZ* under the control of a 1.8-kbp *HAP4* promoter.	This study	Figure 1A
pDC162	YIp356-KGD1-lacZ, a yeast integrative vector encoding a *KGD1-lacZ* reporter gene and a *URA3* section marker.	This study	Figure 2
pDC210	pRS416-HAP4p-HAP4-HA, expressing Hap4 from its native promoter with a 3xHA tag at the C-terminus.	This study	Figures 2B,C, 3B, 4, 5, S1 and S5
pDC216	pRS416-GPDp-HAP4-HA, expressing Hap4 from a *GPD* promoter with a 3xHA tag at the C-terminus.	This study	Figures 2B,C, 3A and S2
pDC218	pRS416-TEF2p-HAP4-HA, expressing Hap4 from a *TEF2* promoter with a 3xHA tag at the C-terminus.	This study	Figure 2B,C
pDC160	pRS416-KGD1-lacZ, expressing *lacZ* under the control of a 696-bp *KGD1* promoter.	This study	Figure 6
pMC106	pRS416-SDH1-lacZ, expressing *lacZ* under the control of a 740-bp *SDH1* promoter.	This study	Figure 6

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
