# Peer review of "Ubiquitin-Conjugating Enzymes Ubc1 and Ubc4 Mediate the Turnover of Hap4, a Master Regulator of Mitochondrial Biogenesis in Saccharomyces cerevisiae"

_microorganisms, 2022, doi:10.3390/microorganisms10122370_

Round 1
Reviewer 1 Report
Capps et al. microorganisms 2068605 peer review
This manuscript is very well written, ist easy to understand and deals with the problem of metabolic regulation in respiratory deficient yeast and in yeast cells in response to a shift in carbon source from fermentative sources (sugars) to respiratory carbon sources (ethanol, acetate and others). Likewise the opposite shift from respiratory to fermentative carbon sources is studied. The paper deals with mitochondrial biogenesis and with the retrograde response.
The paper is technically sound as are the methods used (reporter systems, use of transgenic yeast, immune tags, protein analysis and time course) and the results are clear and in good relation to previously known facts about this metabolic shift. The main finding is this: Hap4, the catalytic component of the Hap2/3/4/5 complex is tightly regulated at the transcriptional and at the protein level (degradation through the proteasome) in the metabolic shift experiments mentioned above. Hap4 levels direct the metabolic shift and the biogenesis of mitochondria.This finding is new and interesting.
Style, language and overall clarity of the manuscript is excellent.
Minor points of critique:
i)in part of the figures the significance of the differences found are indicated by an asterisk without explaining the level of significance. This must be corrected. In another part of the paper significance is explained with p values which is the preferred way to deal with this problem.
ii)in the beginning of the paper the importance of the vacuole for the studied degradation of Hap4 is mentioned (this aspect is not studied in depth in the paper). The importane of the proteasome and ubiquitination system for Hap4 degradation is studied in detail. This system occurs in the cytoplasm and in the nucleus, but not in the vacuole. Perhaps this propblem should be mentioned and discussed in the paper.
Taking into account the aspects mentioned above, my judgement is „minor revision“.
Reviewer 2 Report
The manuscript “Ubiquitin-conjugating enzymes Ubc1 and Ubc4 mediate the turnover of Hap4, a master regulator of mitochondrial biogenesis in Saccharomyces cerevisiae” is interesting and well written. The authors show that loss of mitochondrial DNA increases the instability of Hap4-HA. This is due both to reduced transcription from HAP4 promoter and destabilization of Hap4 proteins. Moreover, Hap4 turnover requires the 26S proteasome and is mediated by Ubc1 and Ubc4 ubiquitin-conjugating enzymes.
The following minor comments can strengthen the manuscript:
1. Figure 3A shows reduced steady state Hap4-HA protein level in p0 cells. Could the authors explain, why the protein levels of Hap4-HA at t=0 in Figure 3B are the same in p0 and p+ cells, although the loading is similar?
2. Figure 4B. The conclusion that Hap4 is ubiquitinated is not supported with enough data. The high-mobility bands could be ubiquitinated Hap4-HA, but additional experiments are needed to prove this. The authors should tune their conclusion or try to do HA-pull-down of Hap4, followed by Western blot with Ubiquitin-antibody and HA-antibody. Alternatively, they can do Ubiquitin-IP and Western blot with HA-antibody to detect Hap4 in the elution fraction.
3. The data in Figure 5C requires quantification, similar to Figure 3C.
4. The authors should avoid “Data not shown" (2 times in the manuscript). They can include the data as Supplementary Material.
